# Parental Origin of the *RB1* Gene Mutations in Families with Low Penetrance Hereditary Retinoblastoma

**DOI:** 10.3390/cancers13205068

**Published:** 2021-10-10

**Authors:** Ekaterina A. Alekseeva, Olga V. Babenko, Valentina M. Kozlova, Tatiana L. Ushakova, Tatiana P. Kazubskaya, Marina V. Nemtsova, Galina G. Chesnokova, Dmitry S. Mikhaylenko, Irina V. Bure, Alexey I. Kalinkin, Ekaterina B. Kuznetsova, Alexander S. Tanas, Sergey I. Kutsev, Dmitry V. Zaletaev, Vladimir V. Strelnikov

**Affiliations:** 1Laboratory of Epigenetics, Research Centre for Medical Genetics, 115522 Moscow, Russia; polyakk@list.ru (O.V.B.); nemtsova_m_v@mail.ru (M.V.N.); galina@epigenetic.ru (G.G.C.); dimserg@mail.ru (D.S.M.); alexeika2@yandex.ru (A.I.K.); kuznetsova.k@bk.ru (E.B.K.); tanas80@gmail.com (A.S.T.); kutsev@mail.ru (S.I.K.); zalnem@mail.ru (D.V.Z.); vstrel@list.ru (V.V.S.); 2Laboratory of Medical Genetics, I.M. Sechenov First Moscow State Medical University (Sechenov University), 119992 Moscow, Russia; bureira@mail.ru; 3Blokhin National Medical Research Center of Oncology, Ministry of Health of the Russian Federation, 115478 Moscow, Russia; valentina-mk2011@yandex.ru (V.M.K.); ushtat07@mail.ru (T.L.U.); kazubskaya@yahoo.com (T.P.K.); 4Russian Medical Academy of Continuous Professional Education, 125993 Moscow, Russia

**Keywords:** hereditary retinoblastoma, *RB1*, penetrance, expressivity, parental origin, low penetrance mutation, NGS

## Abstract

**Simple Summary:**

Some families with hereditary retinoblastoma exhibit mild phenotype with low penetrance and variable expressivity, including complete absence of clinical signs of the disease in some carriers of the germline *RB1* mutation. The identification of low-penetrance mutations in the *RB1* gene and the study of their inheritance in pedigrees is contributing to understanding the mechanisms underlying the development of retinoblastoma with low penetrance. It is important both for further expansion of knowledge in the field of molecular genetics of retinoblastoma, and for competent genetic counseling and subsequent clinical management of families with this form of the disease. Our results support an assumption that parental origin of an *RB1* mutation influences the likelihood of developing retinoblastoma. We also revealed a relatively high frequency of asymptomatic carriage of the *RB1* mutations among the parents of retinoblastoma patients, highlighting the utmost necessity for molecular analysis among the probands’ relatives irrespective of their clinical status and family history of retinoblastoma.

**Abstract:**

Our aim was to identify *RB1* alterations causing hereditary low penetrance retinoblastoma and to evaluate how the parental origin of an *RB1* mutation affects its phenotypic expression. By NGS and MLPA, *RB1* mutations were found in 191 from 332 unrelated retinoblastoma patients. Among patients with identified *RB1* mutations but without clinical family history of retinoblastoma, 7% (12/175) were found to have hereditary disease with one of the parents being an asymptomatic carrier of an *RB1* mutation. Additionally, in two families with retinoblastoma history, mutations were inherited by probands from unaffected parents. Overall, nine probands inherited *RB1* mutations from clinically unaffected fathers and five, from mothers. Yet, we gained explanations of maternal “unaffectedness” in most cases, either as somatic mosaicism or as clinical presentation of retinomas in involution, rendering the proportion of paternal to maternal truly asymptomatic mutation carriers as 9:1 (*p* = 0.005). This observation supports an assumption that parental origin of an *RB1* mutation influences the likelihood of developing retinoblastoma. Additionally, our study revealed a relatively high frequency of asymptomatic carriage of the *RB1* mutations among the parents of retinoblastoma patients, highlighting the utmost necessity of molecular analysis among the probands’ relatives irrespective of their clinical status and family history of retinoblastoma.

## 1. Introduction

Retinoblastoma is the most common cancer affecting the retina in children, with an incidence of 1:16,000–1:18,000, accounting for 3% of all pediatric cancers [1,2]. A tumor develops from the cone precursors, and is characterized by a high degree of malignancy, invasive growth, and rapid metastasis to the neighboring organs and tissues [3].

Retinoblastoma is usually diagnosed in the first two years of a child’s life. The main clinical symptoms of retinoblastoma are leucocoria, strabismus, poor vision, redness of the eye with pain in it, and proptosis. Ophthalmoscopy reveals unifocal or multifocal intraretinal transparent tumor nodes. As the tumor grows, which can occur in a three-dimensional plane, the nodes become opaque, dull white, and vascularized. Additionally, with retinoblastoma, diffuse retinal detachment and tumor seedings into the vitreous body are revealed [3,4]. In approximately 3% of cases, there is a tendency for spontaneous regression of retinoblastoma [5].

Retinoma is diagnosed in both childhood and adulthood. Most patients are diagnosed with retinoma at their visit to an ophthalmologist for an eye examination because of retinoblastoma in their family members, especially first-degree relatives. The diagnosis can also be made in the case of complaints of decreased vision or strabismus. Ophthalmoscopic features of retinoma are translucent gray nodes in the retina, calcification, changes in the pigment epithelium of the retina, and chorioretinal atrophy [6,7]. Retinoma has a wide range of clinical phenotypes. Characteristic symptoms and a positive family history are key points for diagnosis, but only 10% of cases show all four ophthalmoscopic features, (1) semi-transparent retinal tumor (88%), (2) calcification (63%), (3) changes in the retinal pigment epithelium (54%), (4) chorioretinal atrophy (54%) [7]. The presence of chorioretinal atrophy indicates tumor regression.

However, the differential diagnosis of retinoblastoma and retinoma may be challenging. When making a differential diagnosis, the age of the patient is of great importance. In addition, the absence of tumor progression without any treatment speaks in favor of retinoma.

Molecularly development of retinoblastoma is based on biallelic inactivation of the *RB1* tumor suppressor gene [8,9]. Both of the inactivating mutations are somatic in sporadic (noninherited) retinoblastoma, which accounts for 60% of all cases. The noninherited form manifests itself as a unilateral tumor and is usually diagnosed after 2 years of age and, in the majority of cases, before 4 years of age [10].

Hereditary retinoblastoma is diagnosed in 40% of cases. In this form of the disease, a germline mutation of one of the *RB1* alleles predisposes to tumor development and determines familial transmission, while a somatic mutation of the other allele is acquired in utero or in early childhood and triggers tumorigenesis. Familial retinoblastoma manifests itself at an earlier age compared with the sporadic form (the mean age at diagnosis is 12 months) and is multifocal and bilateral in the majority of cases. Carriers of a germline mutation are additionally at higher risk of developing subsequent non-ocular primary cancers [11]. The disease shows an autosomal dominant inheritance and high (above 90%), but not fatal penetrance: there are retinoblastoma families (with two or more carriers of the same germline mutation in the pedigree) that display milder phenotypes with low penetrance (some carriers of the germline mutation do not develop the disease) and variable expressivity (the same mutation is expressed as a unilateral or bilateral disease in different family members) [12,13].

Thus, it is of immense importance to understand the mechanisms that underlie hereditary retinoblastoma with low penetrance in order to improve our knowledge of the molecular genetics of this disease and to provide better genetic counseling and subsequent clinical management.

The hereditary retinoblastoma phenotype is thought to depend on the functional type of the underlying *RB1* mutation [13]. In turn, the molecular mechanisms responsible for the variation in phenotypic expression of one and the same mutation among different family members are currently associated with the parental origin of the *RB1* mutation [13,14].

## 2. Materials and Methods

### 2.1. Clinical Samples

Peripheral blood samples were obtained from 332 unrelated retinoblastoma patients, including 226 with the unilateral and 106 with the bilateral form. The study was conducted in accordance with the Declaration of Helsinki and was approved by the Institutional Ethics Committee of the Research Centre for Medical Genetics. Written informed consent was obtained from each participant involved in the study. In Russia, conservative treatment of retinoblastoma is widely available, thus design of the present study was based on molecular testing of blood samples only.

A family history of retinoblastoma was known for 16 (4.8%) patients, 2 of which had unilateral and 14, bilateral retinoblastoma. In all pedigrees where *RB1* mutations were found in blood cells either by sequencing or MLPA, both parents agreed to give their blood samples and were tested for the carriage of the mutation found in the proband. In clinically familial cases, extra relatives underwent molecular testing provided that they were available and agreed to give their blood samples (families ## 261, 319, 360, 398).

Genomic DNA was isolated from peripheral blood lymphocytes by standard phenol–chloroform extraction.

### 2.2. Mutation Screening by NGS 

Screening for point mutations and small insertions/deletions of the *RB1* gene was performed by high-throughput parallel sequencing using the Ion Torrent platform (Thermo Fisher Scientific, Waltham, MA, USA). The panel of primer pairs for library preparation was designed using AmpliSeq Designer software (Thermo Fisher Scientific, Waltham, MA, USA). Target regions include all coding sequences of the *RB1* gene, adjacent intron regions, and untranslated regions (5′-UTR & 3′-UTR). For library preparation, Ion AmpliSeq Library Kit 2.0 (Thermo Fisher Scientific) was used. The reaction was carried out according to the standard protocol recommended by the manufacturer. Aliquots from the prepared libraries were subjected to clonal emulsion amplification on the Ion OneTouch instruments using the Ion OneTouch 200 Template Kit (Thermo Fisher Scientific, Waltham, MA, USA). Sequencing was performed on the Ion Torrent PGM genomic sequencer using an Ion PGM 200 Sequencing Kit (Thermo Fisher Scientific, Waltham, MA, USA). The results were analysed with Torrent Suite software consisting of Base Caller, Torrent Mapping Alignment Program TMAP, and Torrent Variant Caller. Genetic variants were annotated with ANNOVAR software (Philadelphia, PA, USA) [15]. Visual data analysis, manual filtering of sequencing artefacts and sequence alignment were performed using the Integrative Genomics Viewer (IGV) [16].

### 2.3. Sanger Sequencing 

Once a candidate genetic variant was detected in a proband, DNA samples of the proband’s parents and sibs were tested for this variant by Sanger sequencing of the DNA fragments amplified by PCR from the flanking primers. The direct sequencing of individual PCR products was performed on the automatic genetic analyzer ABI PRISM 3500 (Thermo Fisher Scientific) according to the manufacturer’s protocols.

### 2.4. Multiplex Ligation-Dependent Probe Amplification

Screening for large deletions within or containing the *RB1* gene was performed by MLPA. The MLPA reaction was carried out with a SALSA MLPA P047-D1 RB1 kit according to the standard protocol recommended by the manufacturer (MRC Holland, http://www.mlpa.com) (accessed on 2 August 2021). Capillary electrophoresis of MLPA products was performed on an ABI PRISM 3500 genetic analyzer in accordance with the manufacturer’s instructions. The data obtained were processed using Coffalyser.NET software provided by MRC Holland.

### 2.5. Statistical Analysis

Paternal to maternal truly asymptomatic mutation carrier groups were compared using Fisher’s exact test.

## 3. Results

Among 332 unrelated retinoblastoma cases included in the study, pedigree segregation analysis revealed 16 hereditary retinoblastoma families, including four families (25%) with low penetrance and/or variable expressivity of the disease (families 261, 286, 319, and 360; ordinal numbers are those in the sample collection maintained in our lab). In all families with retinoblastoma history, we identified molecular genetic alterations of the *RB1* gene either by DNA sequencing or by multiplex ligation-dependent probe amplification (MLPA) (Table 1).

Peripheral blood DNA samples of the remaining 316 probands without clinical family history of retinoblastoma were assessed by DNA sequencing and MLPA. Causative genetic variants of the *RB1* gene were identified in 175 cases. The latter underwent familial segregation analysis, and 7% (12/175) were found to have hereditary disease with one of the parents being an asymptomatic carrier of an *RB1* mutation (Table 2).

Taken together, 12 patients that inherited *RB1* mutations in the families without familial history of retinoblastoma (Table 2), and two patients from the families with familial history but without clinical signs of the disease in the probands’ parents (## 261 and 319; Figure 1), constituted a cohort of 14 patients who inherited *RB1* mutations from their clinically asymptomatic mutation carrier parents, nine from fathers and five from mothers (Figure 2). Yet, most cases (4 from 5) of maternal “unaffectedness” gained their explanations after more detailed molecular genetic and clinical examinations.

In family 482, the mother was demonstrated to be mosaic for the mutation (c.887del: wild type, 75%; deletion, 25%, calculated as the mean ratio of the relative amplitudes of the peaks for the available overlapping nucleotide signals; Figure 3) and consequently lacked clinical signs of the disease [17]. It should be pointed out that we calculated a relative ratio of mutant/wild type alleles in blood cells, which is not reflective of what might have occurred in the cells that gave rise to the retina during development (most likely, they arose from non-mutant cells). The ratio in blood cells should not be considered clinically relevant in terms of retinoblastoma in a mosaic mutation carrier. Yet, a mosaic carrier has an increased risk of developing other malignant neoplasms, since a certain percentage of the cells possess the mutant *RB1* allele.

In families 359, 472, and 594, the mothers who were heterozygous carriers of *RB1* nonsense mutations (c.1363C > T, c.1345G > T, and c.2293_2297del, respectively) were found to have retinomas at involution by fundoscopy (Figure 4 and Figure 5). Retinoma is thought to develop in the absence of additional molecular events necessary for its progression to retinoblastoma [18,19]. In the proband’s mother in family 359, ultrasound examination revealed two foci of calcification with chorioretinal dystrophy around them on the retina of the left eye. These findings were interpreted by the oncologist as retinoma foci or spontaneous involution of retinoblastoma at an early age. In family 472, examination of the proband’s mother revealed a focus of calcification with chorioretinal dystrophy around it on the periphery of the retina of the left eye, considered by an oncologist as a retinoma focus or spontaneous involution of retinoblastoma at an early age. The proband’s mother in family 594 presented with congenital bilateral staphyloma, coloboma of the choroid as a consequence of chorioretinitis. In this case, the oncologist’s diagnosis was an intrauterine spontaneous involution of bilateral retinoblastoma.

All the asymptomatic fathers of the probands with retinoblastoma underwent additional examinations, including fundoscopy and ultrasound of the eye, which resulted in no remarkable retinal findings.

Thus, after in-depth molecular and clinical evaluation, we gained explanations of maternal “unaffectedness” in most cases, either as somatic mosaicism or as clinical presentation of retinomas in involution, rendering the proportion of paternal to maternal truly asymptomatic mutation carriers as 9:1 (Figure 2). This difference is statistically significant (*p* = 0.005, Fisher’s exact test).

A total of 15 inherited low penetrance mutations were observed in our cohort of retinoblastoma patients. The mutations led to the disease with incomplete penetrance in 16 families (identical mutations were detected in two families). The probands with retinoblastoma inherited the *RB1* mutation from their fathers who were asymptomatic carriers or had a milder disease form (late-onset and/or unilateral retinoblastoma) in 11 families and from the mothers who were asymptomatic carriers in five families. Additional clinical and molecular genetic tests revealed retinoma at the involution stage in the asymptomatic mothers in three families and mosaicism for the mutation with a low proportion of cells carrying the mutant allele in the mother in one family. Thus, in all cases with unexplained incomplete penetrance, it is paternal inheritance of the mutation that was observed in the majority of families in our cohort (11 families vs. 1 family where the mutation was inherited from the mother).

## 4. Discussion

In this work, molecular genetic assessment of DNA samples from peripheral blood lymphocytes of 332 unrelated retinoblastoma patients resulted in identification of causative *RB1* gene mutations in 191 (58%) of them. Such efficacy of *RB1* germline mutation screening in an overall cohort of retinoblastoma patients is consistent with previous reports [20,21]. Efficacy of *RB1* mutation screening in blood samples depends on the clinical form of retinoblastoma (unilateral or bilateral) and on the family history (inherited or sporadic disease). In our cohort, causative mutations were found in all families with retinoblastoma history. In a group of 316 sporadic retinoblastoma patients without a family history (223 patients with unilateral form of the disease and 93 with bilateral form) mutations in the *RB1* gene in peripheral blood DNA were detected in 55% (175/316) of cases. Among them, 98% (91/93) of patients with bilateral form of the disease demonstrated *RB1* mutations. Such frequency of *RB1* mutations in the bilateral form of the disease is consistent with the results previously reported by other authors [20,21]. Among 223 patients with unilateral form of the disease 84 mutations were found, of which eight were inherited. Thus, in a sporadic unilateral retinoblastoma cohort we detected mutations in 35% (76/215) of cases. We can suggest at least two reasons explaining such a high percentage of germline mutations identified by us in the unilateral form of the retinoblastoma. The first attributes to our sequencing and analysis technique. To search mutations, we used deep NGS, targeting the mean read depth of 400×, and developed an in-house algorithm based on bioinformatic and statistical approaches to reliably identify mosaic mutations with low mutant allele representation. As a result, we identified somatic mosaicism in 8% of cases with unilateral retinoblastoma, and this increases the percentage of mutation detection. Another reason is the lack of follow-up for a number of cases. We as a rule perform DNA testing for patients at a very early stage of disease and register the form of retinoblastoma at that moment, which is unilateral. In a number of patients, the second eye may also become affected later, but, unfortunately, it has not been possible to track the further clinical history of all patients as they are later treated in numerous far-away local centers.

Identification of *RB1* mutations in retinoblastoma patients is possible in about 95% of cases, irrespective of clinical form of the disease and family history, only if the tumor material is available for molecular genetic testing. We reported a 95% efficacy of *RB1* mutation screening in our early studies, when enucleation was an inevitable treatment option and tumor samples were available for each proband under study [9]. In the last two decades, the leading trend in RB treatment is organ preserving therapy [22]. Advances in conservative RB treatment radically reduce the rates of enucleations rendering tumor material unavailable for laboratory assessment. In Russia, conservative treatment of RB is widely available [23], thus design of the present study was based on molecular testing of blood samples only. Yet, such an approach is sufficient to address the key question of the study, assessment of the parental origin of the *RB1* gene mutations in families with low penetrance hereditary retinoblastoma.

The hereditary retinoblastoma phenotype is thought to depend on the type of the germline (first) mutation affecting one of the *RB1* alleles [24,25]. *RB1* mutations are classified into three categories depending on the pRB functional activity. One category includes the mutations that lead to the absence of the *RB1* protein product from the cell and, therefore, loss of its function (nonsense and frameshift mutations). These mutations cause a premature transcription termination and subsequent nonsense-mediated degradation of the defective mRNAs [26]. Patients with germline mutations of this category display complete penetrance and the bilateral form of the disease with multifocal damage to the retina [14]. Mutations of the second category result in a lower amount of normal pRB (mutations at the promoter region or at splice sites) [13]. The third category includes the mutations that partly inactivate pRB (missense mutations and deletions/insertions without a frame shift). These mutations occur in the coding gene region, but do not terminate its transcription prematurely; the mutations determine partial loss of function, for example, by destabilizing the protein or abolishing its additional activities. Mutations of the second and third categories cause retinoblastoma with incomplete penetrance and, usually, fewer tumor foci [13,24].

Of the ten low-penetrance mutations that the retinoblastoma patients inherited from their fathers, five are splice site mutations, three are missense mutations, and two are frameshift deletions (Table 2). Thus, eight mutations fall into the second category and may cause retinoblastoma with incomplete penetrance in accordance with the previously suggested model, and the remaining two may be considered loss of function mutations and attributed to the first category that is not likely to demonstrate incomplete penetrance. Note, however, that both of the potential loss of function frameshift deletions are located in the first exon of the *RB1* gene. We suggest that it is the location in the first exon, abolishing the expression of a longer, naturally weakly expressed transcript and retaining the expression of the shorter one, that provides a possibility for such loss of function mutations to manifest as low penetrant. Modulation of penetrance of the disease caused by frameshift mutations may also be achieved by internal translation initiation. Sanchez et al. (2007) reported a family with a low penetrance *RB1* mutation comprising a 23-basepair duplication in the first exon of *RB1* (c.43_65dup) producing a frameshift in exon 1 and premature chain termination in exon 2. The authors demonstrated that this mutation did not cause appreciable NMD, and transcript expression in tissue culture cells and translation in vitro revealed that alternative in-frame translation start sites involving Met113 and possibly Met233 were used to generate truncated RB1 products (pRB94 and pRB80), known and suspected to exhibit tumor suppressor activity [27].

An effect of the parental origin of the *RB1* mutation is currently thought to provide a molecular mechanism that underlies the variation in phenotypic expression of the same mutation in different members of a family with hereditary retinoblastoma [12,14,20].

The *RB1* gene is known to harbor a 1.2-kb imprinted region presented by a CpG island (CpG 85) in intron 2 that shows differential methylation depending on the parental origin of the allele; i.e., the region is methylated in the maternal chromosome and nonmethylated in the paternal one. Two other CpG islands, CpG 106, and CpG 42, reside in the *RB1* gene. The island CpG 106 includes the promoter and exon 1 and is characterized by biallelic lack of methylation, thus allowing expression of the major pRB-coding transcript from both *RB1* alleles. The island CpG 42 is in intron 2, is methylated in both chromosomes, and lacks regulatory activity [14,20].

There is evidence that CpG 85 is part of a 5′-truncated processed pseudogene that originates from the *PPP1R26P1* protein-coding gene, which is in chromosome 9 and is integrated in *RB1* in the inverse orientation. CpG 85 acts as a promoter for an alternative *RB1* transcript, which is expressed only from the non-methylated paternal chromosome. In addition, although the total expression level of mRNA transcripts synthesized from the paternal allele may be expected to be higher than from the maternal one, expression from the paternal allele is actually two to three times lower because transcriptional interference arises when both regular and alternative transcripts are expressed [14,20].

Demethylation of CpG 85 in lymphoblast cell lines treated with the demethylating agent 5-aza-2’-deoxycytidine has been observed to result in equal levels of mRNA expression from the two *RB1* alleles because the expression profile of the maternal allele becomes similar to that of the paternal one [14].

Mice have not been observed to have imbalanced levels of *RB1* expression from the paternal and maternal alleles. The mouse *RB1* gene lacks a CpG island homologous to human CpG 85. The observation indicates that differentially methylated CpG 85 is responsible for biased allelic expression of human *RB1* [14].

It is currently thought that transcriptional interference is responsible for lower *RB1* expression from the paternal allele compared with the maternal one. Transcriptional interference is a mechanism where transcription of one gene suppresses transcription of another [20]. A transcription complex that binds to non-methylated CpG 85 of the paternal chromosome to synthesize the alternative *RB1* transcript, blocks the progress of the transcription complex that synthesizes the major transcript from the same allele, thus decreasing the total expression level from the paternal chromosome. Thus, the above data demonstrate that *RB1* expression is epigenetically regulated depending on the parental origin of the allele [14,20].

Other studies have confirmed that the sex of the transmitter of an *RB1* mutation correlates with disease penetrance [12,28,29]. According to published data, low-penetrance *RB1* mutations inherited from the father more often cause retinoblastoma and determine its more severe form than the same mutations inherited from the mother. Tumor suppressor activity of pRB is thought to be high enough to prevent retinoblastoma when a low-penetrance mutation is inherited maternally. In contrast, when a mutant allele is transmitted from the father, low residual activity of the protein results from its lower expression and mimics a loss-of-function mutation, leading to retinoblastoma [8].

Although several reports have already shown evidence of a parent-of-origin effect in retinoblastoma presentation, an intriguing question is whether all cases of incomplete *RB1* mutation penetrance may be explained by this effect, or whether the search has to be undertaken for other molecular mechanisms, such as, for example, impact of genetic variants outside the *RB1* gene (“genomic environment”). One way to address this question is to accumulate observations of different *RB1* low penetrance mutations in well characterized trios. Observation that males constitute an overwhelming number of truly asymptomatic *RB1* mutation carriers may be interpreted as evidence that a parent-of-origin effect is a principal, if not a sole explanation for the incomplete penetrance of the *RB1* gene mutations. Our present study is one that fully supports this conclusion. It is of note however, that the numbers of non-penetrant transmitting parents reported here are small, but this is a consequence of the low frequency of asymptomatic carriage of retinoblastoma in the population. The data of course should not be over interpreted: more substantiated conclusions may be made at least in the meta-analyses based on studies like ours, when the number of such increases.

## 5. Conclusions

In this study of low penetrance hereditary retinoblastoma, we described a significant group of patients without family history of retinoblastoma, that is, all patients were admitted as cases with sporadic disease. In the previously published articles on low penetrance hereditary retinoblastoma, in most cases, familial cases were described. Additionally, we demonstrated new mutations in the *RB1* gene that should be classified as low penetrance mutations, that were not previously reported, and support low penetrance for three previously reported ones, c.607+1G > T [14,22,28], c.1981C > T [12], and c.939G > A [30]. Our study revealed a relatively high frequency of asymptomatic carriage of the *RB1* mutations among the parents of retinoblastoma patients, highlighting the utmost necessity for molecular analysis among the probands’ relatives irrespective of their clinical status and family history of retinoblastoma. Knowing the relatives *RB1* status does not affect treatment and further clinical management of the proband. However, the identification of asymptomatic carriers of the *RB1* gene mutation in a family is important for genetic counseling of a family in terms of further childbirth, since carriers of *RB1* mutations with low penetrance have an increased risk of having a child with retinoblastoma compared to the general population. In addition, carriage of a low penetrance mutation in the *RB1* gene increases the risk of developing other malignant neoplasms in various locations throughout life [21,29].

## Figures and Tables

**Figure 1 cancers-13-05068-f001:**
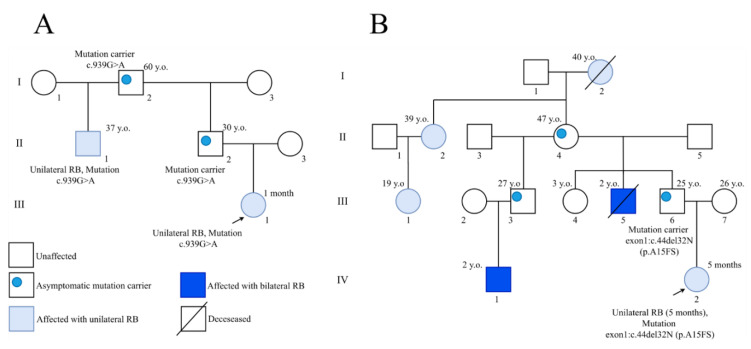
Pedigrees (261, **A**, and 319, **B**) with familial retinoblastoma history but without clinical signs of the disease in the probands’ parents.

**Figure 2 cancers-13-05068-f002:**
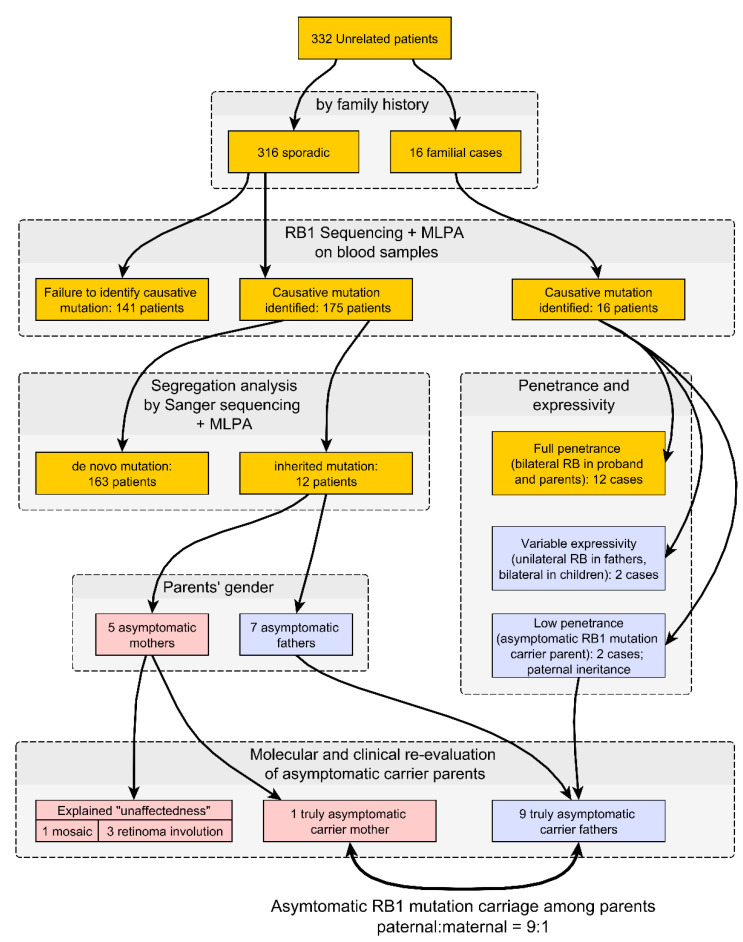
Flowchart of the study, summarizing cohort information, methods, results, and conclusions. Genomic DNA from peripheral blood lymphocytes of 332 unrelated retinoblastoma patients (316 sporadic and 16 familial cases) was assessed for the *RB1* gene alterations by sequencing and multiplex ligation-dependent probe amplification (MLPA). *RB1* point mutations or gross deletions were identified in all familial cases, as well as in 175 of 316 cases deemed to be sporadic by clinical evaluation. Among these 175 cases, segregation analysis revealed the inherited nature of probands’ RB1 mutations in 12 families, with 5 cases of inheritance from asymptomatic mothers and 7 from asymptomatic fathers. Taken together, these 12 patients that inherited *RB1* mutations in the families without familial history of retinoblastoma, and 2 patients from the families with familial history but without clinical signs of the disease in the probands’ parents, constituted a cohort of 14 patients who inherited *RB1* mutations from their clinically asymptomatic parents, 9 from fathers and 5 from mothers. Further clinical and molecular re-evaluation of asymptomatic carrier parents revealed 1 mosaic mutation carrier mother and 3 mothers with retinoma in involution, rendering the proportion of paternal to maternal truly asymptomatic mutation carriers as 9:1.

**Figure 3 cancers-13-05068-f003:**
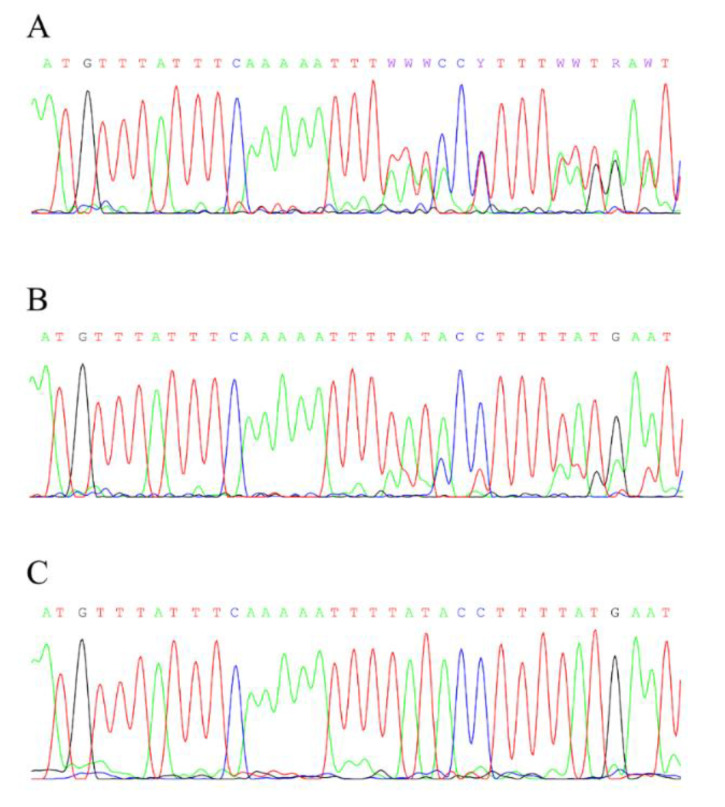
Sanger sequences demonstrating *RB1* c.887del mosaicism in the peripheral blood lymphocytes DNA of an asymptomatic mutation carrier mother of proband 482. (**A**), proband; (**B**), mother; (**C**), normal control.

**Figure 4 cancers-13-05068-f004:**
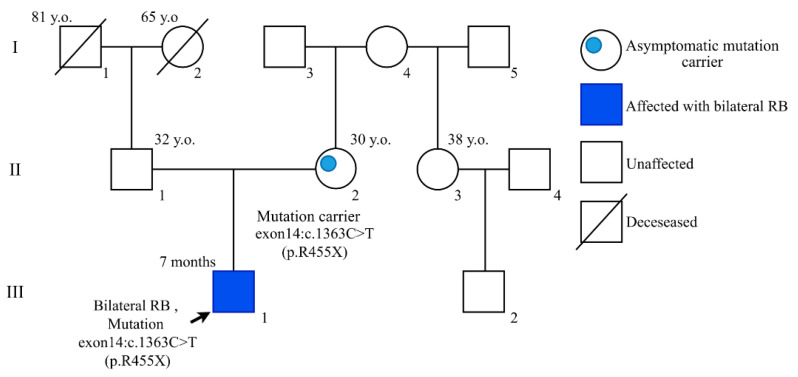
Pedigree (#359) with familial retinoblastoma history but without clinical signs of the disease in the probands’ parents revealed at initial visit. Further clinical re-evaluation by fundoscopy revealed retinoma at involution in the proband’s mother (see Figure 5).

**Figure 5 cancers-13-05068-f005:**
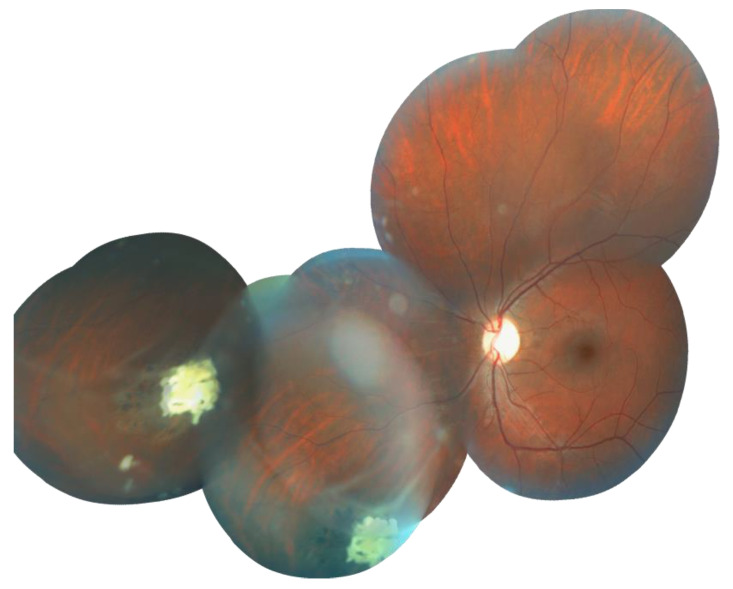
Picture of the fundoscopy performed for the clinically asymptomatic mutation carrier mother (II-2) of proband (III-1) from family #359. Retinoma at involution. The foci of calcification and foci of chorioretinal dystrophy around them creeping onto the retina.

**Table 1 cancers-13-05068-t001:** Spectrum of *RB1* alterations observed in patients with family history of retinoblastoma.

Family #	Mutation	Mutation Type	Location	RB Clinical form in the Proband	Other Mutation Carriers in the Family/Retinoblastoma Form
164	c.32_63del	Frameshift	Exon 1	Bilateral	Mother/bilateral
261	c.939G > A	Splice site (missense splice)	Exon 9	Unilateral	Father, asymptomatic carrier of the mutation Father’s half-sib/unilateral Grandfather/asymptomatic carrier
286	c.380+1G > A	Splice site	Exon 3	Bilateral	Father/unilateral
306	c.1072C > T	Nonsense	Exon 11	Bilateral	Father/bilateral
319	c.45_76del	Frameshift	Exon 1	Unilateral	Father, asymptomatic carrier of the mutation Father’s half-sib/unilateral Many cases in the paternal lineage
323	c.958C > T	Nonsense	Exon 10	Bilateral	Father/bilateral
327	c.958C > T	Nonsense	Exon 10	Bilateral	Mother/bilateral
347	c.54_76del	Frameshift	Exon 1	Bilateral	Father/bilateral
360	c.1696-2A > G	Splice site	Intron 17	Bilateral	Father/unilateral Father’s sib/unknown
372	c.1735C > T	Nonsense	Exon 18	Bilateral	Mother/bilateral
388	NC_000013.11:g. (43412928_48258929)_ (48381391_48453040)del	Gross deletion	Exons 1–27	Bilateral	Mother/bilateral
394	c.1654C > T	Nonsense	Exon 17	Bilateral	Father/bilateral
398	c.1724del	Frameshift	Exon 18	Bilateral	Father/bilateral Sib/bilateral
409	c.608-12T > G	Splice site	Intron 6	Bilateral	Mother/bilateral
538	NC_000013.11:g. (48463554_48465087)_ (48465224_48473094)del	Intragenic deletion	Exons 22–23	Bilateral	Mother/bilateral
539	c.1233_1254 ins TAAAGAACTGC ACAGTGAATCC	Frameshift	Exon 13	Bilateral	Father/bilateral

#—ordinal numbers in the sample collection maintained in our lab.

**Table 2 cancers-13-05068-t002:** *RB1* gene mutations inherited by retinoblastoma patients from their clinically asymptomatic parents.

Family #	Mutation	Mutation Type	Location	Retinoblastoma Form in the Proband	Parent—Asymptomatic Mutation Carrier
Families with Purely Asymptomatic Mutation Carrier Parents					
319	c.45_76del	Frameshift	Exon 1	Unilateral	Father
485	c.83del	Frameshift	Exon 1	Unilateral	Father
393	c.607+1G > T	Splice site	Intron 6	Unilateral	Father
533	c.607+1G > T	Splice site	Intron 6	Unilateral	Father
261	c.939G > A	Splice site (missense splice)	Exon 9	Unilateral	Father
255	c. 1364G > C	Missense	Exon 14	Unilateral	Father
566	c.1573G > A	Missense	Exon 17	Unilateral	Father
437	c.1695+5G > T	Splice site	Intron 17	Bilateral	Father
424	c.1981C > T	Missense	Exon 20	Unilateral	Father
522	c.861G > C	Splice site (missense splice)	Exon 8	Bilateral	Mother
Families with asymptomatic mutation carriage in parents explained by either incomplete penetrance or mosaic mutation carriage					
482	c.887del	Frameshift	Exon 9	Bilateral	Mother (mosaic)
472	c.1345G > T	Nonsense	Exon 14	Bilateral	Mother (retinoma involution)
359	c.1363C > T	Nonsense	Exon 14	Bilateral	Mother (retinoma involution)
594	c.2293_2297 del	Frameshift	Exon 22	Bilateral	Mother (retinoma involution)

#—ordinal numbers in the sample collection maintained in our lab.

## Data Availability

Raw sequence reads data are available at the NCBI BioProject, Accession: PRJNA586849, ID: 586849 (https://www.ncbi.nlm.nih.gov/bioproject/586849) (accessed on 2 August 2021).

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
