# Peer review of "Parental Origin of the RB1 Gene Mutations in Families with Low Penetrance Hereditary Retinoblastoma"

_cancers, 2021, doi:10.3390/cancers13205068_

Round 1

Reviewer 1 Report

General observations and critique on the data

The report and data included is interesting but not especially novel following on from reports including Klutz 2002, Eloy 2016 and Imperatore 2018.

It also appears there is a degree of conflation of the phenotypic observatons in different generations which potentially have different somatic and germline aetiologies and therefore a linked interpretation  cannot be fully substantiated

There are inconsistencies internally within the papers figures and more widely reported literature data.

For example text lines 102/103 describe 3 unilateral and 13 bilateral cases in familial cases whereas in table 1 these are listed as 2 and 14.

As far as I could extrapolate from the figures provided the detection of germline RB1 mutations in non familial cases that were unilateral was 83/.224 individuals (37%). This is extremely high and needs an explanation especially as the overall rate of unilateral to bilateral cases was >68% to <32% which is also at the highest end of the range reported in most literature series.

The numbers of non penetrant transmitting parents are small 9 paternal and 5 maternal and therefore without further biological evidence of the underlying differences should not be over interpreted.

The authors in table 2 stress the significance of 9 "unaffected" paternal carriers and only 5 maternal carriers who they emphasis are materially different in 4 cases due to mosaicism or retinomas. There is no mention on whether the ophthalmologists were blinded to which parents were gene carriers or not. Retinal changes can be subtle and difficult to interpret and unconcsious bias with the interpretation cannot be excluded. It is also notable that 8/9 affected children of paternal transmission had unilateral disease and mutations with mutations typically associated with lower levels of penetrance and expressivity, whereas the maternal mutations were all associated with bilateral disease and in 4 cases mutations associated with greater penetrance and expressivity. This alone might explain the parental phenotype observations .Furthermore the clinical presentation in the parent is not only going to be influenced by the nature of the RB1 gene mutation but also on the parent of origin. Without this grandparental data or specific data on the methylation and or gene expression great caution should be placed on the parental genotype phenotype interpretation and assuming it has the same biological mechanism seen in their children. For example the conclusion in lines 330 and 333 conflates parental and child interpretation without data to support this. The apparent excess of "asymptomatic males" should not be assumed to be a reflection of parent of origin effect when there are very different mutations in the paternal and maternally inherited cohorts.

The conclusions about the need for testing unaffected family members for known mutations and the potential greater penetrance and expressivity with certain RB1 gene mutations are important observations to emphasise for the general medical community but are not novel findings.

Specific points on the text which would also benefit from some editorial input

The authors should make clearer they are testing WBC DNA for potential germline mutations (for example line 104) as some readers may confuse the pathway where somatic tumour mutations would be tested first if a tumour sample was available.

Line 83/84 terminology is rather awkward - more common description would be "subsequent non-ocular primary cancers"

Line 84-89 This could be re-written The authors state the penetrance is >90% i.e not 100%, therefore there will by definition be cases where a parent will be non penetrant. The paragraph should be written so the fact explains this observation rather than being written as 2 separate unrelated observations.

Line 139 "Family survey" is rather non specificand an  alternative terms such as "pedigree segregation analysis" or even "obtaining a family history" is easier to understand exactly what was done.

Table 1  It would be preferable that more precise breakpoints are given for the large deletions. Also family 538 with a deletion of exon 22-23 is an intragenic deletion not a gross deletion.

Figure 5 is fundoscopy not an USS

Author Response

Dear reviewer,

Thank you for your valuable observations and critique.

We have addressed your comments as follows:

  • The report and data included is interesting but not especially novel following on from reports including Klutz 2002, Eloy 2016 and Imperatore 2018.

We agree that the topic is not new; however, in this study of low penetrance hereditary retinoblastoma, we are describing for the first time a significant group of patients without family history of the disease, that is, all patients were admitted as cases with sporadic retinoblastoma. In other articles on low penetrance hereditary retinoblastoma, in most cases, familial cases are described. Additionally, we demonstrate new mutations in the RB1 gene that should be classified as low penetrance mutations, that were not previously reported, and support low penetrance for three previously reported ones, с.607+1G˃T, с.1981C˃T, and c.939G˃A. We have added these sentences to “Conclusions”.

  • It also appears there is a degree of conflation of the phenotypic observatons in different generations which potentially have different somatic and germline aetiologies and therefore a linked interpretation cannot be fully substantiated.

It is believed that the phenotypic manifestation of hereditary retinoblastoma depends on the functional type of the first (germline / predisposing), and not the second (somatic) mutation [Buiting K., 2010]. By Sanger sequencing, we have identified the same mutations in the parents as in probands. Only in one case carrier parent had a mosaic mutation, all other asymptomatic carriers were true heterozygotes in white blood cells DNA. In these parents, mutations may be either de novo or inherited, but judging by true heterozygosity in white blood cells, we can assume that the mutant allele is present in any cell of the body with high probability. Based on this, we consider it reasonable to compare different generations of carriers of the RB1 gene mutation with each other.

  • There are inconsistencies internally within the papers figures and more widely reported literature data. For example text lines 102/103 describe 3 unilateral and 13 bilateral cases in familial cases whereas in table 1 these are listed as 2 and 14.

Thank you, we have corrected this in the text.

We have also corrected figure 4 where mother was erroneously marked as unilateral, and improved figure 2 a little to assist readers in understanding our data.

As for inconsistencies within the papers figures and more widely reported literature data, please see our explanations below.

  • As far as I could extrapolate from the figures provided the detection of germline RB1 mutations in non familial cases that were unilateral was 83/.224 individuals (37%). This is extremely high and needs an explanation especially as the overall rate of unilateral to bilateral cases was >68% to <32% which is also at the highest end of the range reported in most literature series..

Among 223 patients with unilateral form of the disease 84 mutations were found, of which 8 were inherited. Thus, in a sporadic unilateral retinoblastoma cohort we have detected mutations in 35% (76/215) of cases. We can suggest at least two reasons explaining such a high percentage of germline mutations identified by us in unilateral form of retinoblastoma. The first attributes to our sequencing and analysis technique. To search mutations, we used deep NGS targeting the mean read depth of 400x, and developed an in-house algorithm based on bioinformatic and statistical approaches to reliably identify mosaic mutations with low mutant allele representation. As a result, we identify somatic mosaicism in 8% of cases with unilateral retinoblastoma, and this increases the percentage of mutation detection. Another reason is the lack of follow-up for a number of cases. We as a rule perform DNA testing for patients at a very early stage of disease and register the form of retinoblastoma at the moment, which is unilateral. In a number of patients, the second eye may also become affected later, but, unfortunately, it is not possible to track the further clinical history of all patients as they are later treated in numerous far-away local centers.  

You can now find these explanations in the first paragraph of the Discussion.

  • The numbers of non penetrant transmitting parents are small 9 paternal and 5 maternal and therefore without further biological evidence of the underlying differences should not be over interpreted.

The numbers are indeed small, but this is a consequence of the low frequency of asymptomatic carriage of retinoblastoma in the population. The data of course should not be over interpreted: more substantiated conclusions may be made at least in the meta-analyses based on the studies like ours, when the number of such will be significant. We have added these to Discussion as final sentences.

  • The authors in table 2 stress the significance of 9 "unaffected" paternal carriers and only 5 maternal carriers who they emphasis are materially different in 4 cases due to mosaicism or retinomas. There is no mention on whether the ophthalmologists were blinded to which parents were gene carriers or not. Retinal changes can be subtle and difficult to interpret and unconcsious bias with the interpretation cannot be excluded.

Additional examinations, including ultrasound and fundoscopy, were obligatorily carried out when a mutation in the RB1 gene was found in one of the parents in the absence of clinical signs of retinoblastoma. Although the ophthalmologists were not blinded, they were motivated to perform careful examination of the fundus and to report subtle changes.

  • It is also notable that 8/9 affected children of paternal transmission had unilateral disease and mutations with mutations typically associated with lower levels of penetrance and expressivity, whereas the maternal mutations were all associated with bilateral disease and in 4 cases mutations associated with greater penetrance and expressivity.

All mutations in our study inherited from “truly” asymptomatic carrier fathers led to the development of a more severe phenotype in children then in fathers. We excluded 4 cases of mothers with mosaicism or retinoma from the analysis, since the mutations inherited from them are full-penetrant and lead to the development of the bilateral form of the disease in most cases. We see no collision here, as far as these were excluded from further analysis.

  • This alone might explain the parental phenotype observations .Furthermore the clinical presentation in the parent is not only going to be influenced by the nature of the RB1 gene mutation but also on the parent of origin. Without this grandparental data or specific data on the methylation and or gene expression great caution should be placed on the parental genotype phenotype interpretation and assuming it has the same biological mechanism seen in their children. For example the conclusion in lines 330 and 333 conflates parental and child interpretation without data to support this. The apparent excess of "asymptomatic males" should not be assumed to be a reflection of parent of origin effect when there are very different mutations in the paternal and maternally inherited cohorts.

As far as most maternally inherited cases were excluded from analysis for being not truly asymptomatic in mothers, leaving only 1 case, it is impossible to draw conclusions on the effect of the biological function of a mutation. Substantiated conclusions will be possible when more cohorts are studied and published.

As for former lines 330-333, we have changed “should be interpreted” to “may be interpreted”, and added limitations of our study at the end of Discussion.

  • The conclusions about the need for testing unaffected family members for known mutations and the potential greater penetrance and expressivity with certain RB1 gene mutations are important observations to emphasise for the general medical community but are not novel findings.

We agree that though these findings in general are not novel, but important observations to emphasize for the general medical community.

  • Specific points on the text which would also benefit from some editorial input
  • The authors should make clearer they are testing WBC DNA for potential germline mutations (for example line 104) as some readers may confuse the pathway where somatic tumour mutations would be tested first if a tumour sample was available.

In Russia, conservative treatment of retinoblastoma is widely available, thus design of a present study was based on molecular testing of blood samples only. We have discussed appropriateness of such design for this study in the Discussion. It is also indicated on the flowchart of the study. In order to make it clearer from the beginning, we have now added this to the “Clinical samples” subsection of Materials and Methods.

  • Line 83/84 terminology is rather awkward - more common description would be "subsequent non-ocular primary cancers"

Fixed.

  • Line 84-89 This could be re-written The authors state the penetrance is >90% i.e not 100%, therefore there will by definition be cases where a parent will be non penetrant. The paragraph should be written so the fact explains this observation rather than being written as 2 separate unrelated observations.

We have tried to rewrite this paragraph accordingly.

  • Line 139 "Family survey" is rather non specificand an alternative terms such as "pedigree segregation analysis" or even "obtaining a family history" is easier to understand exactly what was done.

Fixed.

  • Table 1 It would be preferable that more precise breakpoints are given for the large deletions.

These deletions were identified by MLPA, and are annotated according to current Human Gene Variation Society (HGVS) Recommendations. Such annotation indicates that the deletion break point has not been sequenced, and more precise breakpoints cannot be given.

  • Also family 538 with a deletion of exon 22-23 is an intragenic deletion not a gross deletion.

Fixed.

  • Figure 5 is fundoscopy not an USS.

Fixed.

Reviewer 2 Report

Report on manuscript cancers-1366135

The manuscript is interesting, well presented and clearly written. The aim is to identify the carriers of mutations in the RB1 gene without clinical symptoms, who can transmit the mutation to their offspring, which can give rise to the development of retinoblastoma. The authors point out that it is important to identify these asymptomatic carriers to know the risk in their offspring,

Some errors and some interpretation considerations for this manuscript are as follows:

1) Page 2 lines 74-75: Sporadic retinoblastoma may be unilateral or bilateral not only unilateral. The term sporadic is opposed to familial.

2) Page 2 line 80: “Hereditary retinoblastoma 80 manifests itself at an earlier age as compared with the sporadic form” Hereditary retinoblastoma may be sporadic or familial.

3) Table 1, family 319: “Father, asymptomatic carrier of the mutation • Father’s half-sib / unilateral” According to the pedigree in figure 1 the asymptomatic father had one asymptomatic sib and one bilateral sib already deceased and one unilateral cousin.

4) In this pedigree (319) there is an asymptomatic mother (II-4) who had 2 asymptomatic sons (III-3 and III-6) and 1 bilateral already deceased son (III-5). This asymptomatic mother was not considered in the count of asymptomatic mothers.

5) Page 8 lines 188-190: “In families 359, 472, and 594, the mothers who were heterozygous carriers of RB1 188 nonsense mutations were found 189 to have retinomas at involution”. One of these mothers (pedigree 359) is presented in figure 4 as unilateral retinoblastoma.

6) Page 10 line 212: “maternal “unaffectedness” in most cases, either as somatic mosaicism or as clinical presentation of retinomas in involution” The same consideration as in 5) one of this mother is unilateral in figure 4.

7) Figure 4 and legend to figure 5: “asymptomatic mutation carrier mother (II-2) of proband (III-1) from family #359: Retinoma at involution”. The same consideration as in 5) and 6): this mother is presented in figure 4 as unilateral.

8) Page 10 lines 214 and 226:  The number of low penetrance cases is small (10), not enough to assure that “it is paternal inheritance of the mutation that was observed in the majority of low penetrance families”

9) Mutations identified in these low penetrance families are: 5 splice-site mutations, 3 missense mutations and 2 frameshift mutations. In several reports it is stated that most of low penetrance mutations are promoter mutations, missense mutations and in-frame deletions (references 12 and 13 of the manuscript). Splice-site mutations may be low penetrance in some cases ( for example mutation in the last nucleotide of an exon), but mainly they are high penetrance mutations.

10) The fact of the presence of a frameshift mutation (considered of high penetrance) in exon 1 of the gene, in two low penetrance families (319 and 485) is explained by the authors on the basis of the existence of a second promoter in that exon where the mRNA transcription can be started This is only a hypothesis for explanation of the low penetrance of this mutation. Interestingly, in family 319 there were bilateral cases of retinoblastoma that are not common in low penetrance retinoblastoma.

11) It would be interesting to calculate the ratio of diseased eyes with respect to the carriers of the mutation, which in high penetrance cases is 2 or close to 2, while in low penetrance cases it is lower, between 1 and 1.5 approximately.

Author Response

Dear reviewer,

Thank you for your valuable observations and critique.

We have addressed your comments as follows:

1) Page 2 lines 74-75: Sporadic retinoblastoma may be unilateral or bilateral not only unilateral. The term sporadic is opposed to familial.

Here we have substituted “sporadic” by “noninherited”.

2) Page 2 line 80: “Hereditary retinoblastoma 80 manifests itself at an earlier age as compared with the sporadic form” Hereditary retinoblastoma may be sporadic or familial.

We have changed this to: “Familial retinoblastoma manifests itself at an earlier age…”

3) Table 1, family 319: “Father, asymptomatic carrier of the mutation • Father’s half-sib / unilateral” According to the pedigree in figure 1 the asymptomatic father had one asymptomatic sib and one bilateral sib already deceased and one unilateral cousin.

In order not to overweight the table, we have only indicated “ •   Many cases in the paternal lineage”, and the details are indeed shown on figure 1.

4) In this pedigree (319) there is an asymptomatic mother (II-4) who had 2 asymptomatic sons (III-3 and III-6) and 1 bilateral already deceased son (III-5). This asymptomatic mother was not considered in the count of asymptomatic mothers.

We have not included this asymptomatic mother because she has not undergone clinical re-examination, and thus we cannot classify her case as true unaffectedness.

5) Page 8 lines 188-190: “In families 359, 472, and 594, the mothers who were heterozygous carriers of RB1 188 nonsense mutations were found 189 to have retinomas at involution”. One of these mothers (pedigree 359) is presented in figure 4 as unilateral retinoblastoma.

There was a mistake on figure 4. We have corrected the figure, thank you!

6) Page 10 line 212: “maternal “unaffectedness” in most cases, either as somatic mosaicism or as clinical presentation of retinomas in involution” The same consideration as in 5) one of this mother is unilateral in figure 4.

This mother is correctly listed as asymptomatic, retinoma involution, in table 2. Figure 4 is now in correspondence with this.

7) Figure 4 and legend to figure 5: “asymptomatic mutation carrier mother (II-2) of proband (III-1) from family #359: Retinoma at involution”. The same consideration as in 5) and 6): this mother is presented in figure 4 as unilateral.

Fixed.

8) Page 10 lines 214 and 226:  The number of low penetrance cases is small (10), not enough to assure that “it is paternal inheritance of the mutation that was observed in the majority of low penetrance families”

In order to emphasize that this relates only to the observation made in this very study, we have added “…in the majority of families in our cohort”.

The numbers are indeed small, but this is a consequence of the low frequency of asymptomatic carriage of retinoblastoma in the population. The data of course should not be over interpreted: more substantiated conclusions may be made at least in the meta-analyses based on the studies like ours, when the number of such will be significant. We have added these to Discussion as final sentences.

9) Mutations identified in these low penetrance families are: 5 splice-site mutations, 3 missense mutations and 2 frameshift mutations. In several reports it is stated that most of low penetrance mutations are promoter mutations, missense mutations and in-frame deletions (references 12 and 13 of the manuscript). Splice-site mutations may be low penetrance in some cases ( for example mutation in the last nucleotide of an exon), but mainly they are high penetrance mutations.

Splice site mutations, according to Harbor J.W. 2001, belong to the class of variations that may lead to a decrease of the normal RB protein expression, which, in turn, can lead to the development of low penetrance retinoblastoma. In other studies (Eloy P., 2016; Imperatore V., 2018), various low penetrance splice site mutations with substitutions located in intron regions are described. In addition, one of the mutations from our study, c.607+1G-T (found in two families) was previously described by Klutz M., 2002, as low-penetrant.

10) The fact of the presence of a frameshift mutation (considered of high penetrance) in exon 1 of the gene, in two low penetrance families (319 and 485) is explained by the authors on the basis of the existence of a second promoter in that exon where the mRNA transcription can be started This is only a hypothesis for explanation of the low penetrance of this mutation. Interestingly, in family 319 there were bilateral cases of retinoblastoma that are not common in low penetrance retinoblastoma.

Low penetrance mutations tend to actually lead to the development of a unilateral form of the disease, but bilateral cases have also been described. In the study of Eloy P., 2016, families with low penetrant retinoblastoma, in which cases of the bilateral phenotype are encountered, are described.

11) It would be interesting to calculate the ratio of diseased eyes with respect to the carriers of the mutation, which in high penetrance cases is 2 or close to 2, while in low penetrance cases it is lower, between 1 and 1.5 approximately.

It would be interesting indeed, but most of our pedigrees are trios, and thus such calculation on our cohort unfortunately lacks sense.

Reviewer 3 Report

This is an interesting study which adds evidence to the current theory of low penetrance retinoblastoma mechanism.  It needs minor revision and explanation of mutation numbers found. 

1- The last paragraph of the results section (lines 216-227) is difficult to understand. It seems that the figures for inherited mutations from family history positive and negative families have been confused.  There are 16 families with family history (15 different mutations) and there are additional 12 cases (11 different mutations) where the mutation was found in an asymptomatic parent.  The 16 family history cases do not all have incomplete penetrance (line 218).  This paragraph needs revision. 

2- The percentage of germline mutations identified in the 223 unilateral sporadic cases is not given in the paper.  From the other figures given, it seems that  84/223 (37.7%) exhibited germline mutations.  6 of these were really inherited mutations so the real figure seems to be 78/217 which is 35.9%.  This is an extremely high germline mutation presence in unilateral sporadic rb.  The paper should have a breakdown of figures and an explanation of this very high percentage.

Author Response

Dear reviewer,

Thank you for your general positive perception of our manuscript.

We have addressed your comments as follows:

1- The last paragraph of the results section (lines 216-227) is difficult to understand. It seems that the figures for inherited mutations from family history positive and negative families have been confused.  There are 16 families with family history (15 different mutations) and there are additional 12 cases (11 different mutations) where the mutation was found in an asymptomatic parent.  The 16 family history cases do not all have incomplete penetrance (line 218).  This paragraph needs revision.

You have understood this paragraph correctly. To make it clearer, we have added “A total of 15 inherited low penetrance mutations were observed…” to the first sentence in this paragraph.

2- The percentage of germline mutations identified in the 223 unilateral sporadic cases is not given in the paper.  From the other figures given, it seems that  84/223 (37.7%) exhibited germline mutations.  6 of these were really inherited mutations so the real figure seems to be 78/217 which is 35.9%.  This is an extremely high germline mutation presence in unilateral sporadic rb.  The paper should have a breakdown of figures and an explanation of this very high percentage.

Among 223 patients with unilateral form of the disease 84 mutations were found, of which 8 were inherited. Thus, in a sporadic unilateral retinoblastoma cohort we have detected mutations in 35% (76/215) of cases. We can suggest at least two reasons explaining such a high percentage of germline mutations identified by us in unilateral form of retinoblastoma. The first attributes to our sequencing and analysis technique. To search mutations, we used deep NGS targeting the mean read depth of 400x, and developed an in-house algorithm based on bioinformatic and statistical approaches to reliably identify mosaic mutations with low mutant allele representation. As a result, we identify somatic mosaicism in 8% of cases with unilateral retinoblastoma, and this increases the percentage of mutation detection. Another reason is the lack of follow-up for a number of cases. We as a rule perform DNA testing for patients at a very early stage of disease and register the form of retinoblastoma at the moment, which is unilateral. In a number of patients, the second eye may also become affected later, but, unfortunately, it is not possible to track the further clinical history of all patients as they are later treated in numerous far-away local centers. 

You can now find these explanations in the first paragraph of the Discussion.

Reviewer 4 Report

In the study of Alekseeva et al. entitled “Parental origin of the RB1 gene mutations in families with low penetrance hereditary retinoblastoma” the authors aimed to identify low-penetrance mutations in retinoblastoma and to define the parental origin of the mutations.

The RB1 mutation status was defined using genomic DNA isolated from blood samples of 332 patients with retinoblastoma and parents and other relatives, if available. Mutation analysis was performed using Ion Torrent NGS sequencing, validation by Sanger sequencing and MLPA for the detection of large deletions.

16 patients had a known family history of retinoblastoma and accordingly all carried a constitutional mutation of RB1 in blood DNA. In two families, one parent also carried the mutation in RB1 but did not develop retinoblastoma (= RB1 variant with low penetrance).

Of the 316 cases with no family history, 141 did not show an RB1 mutation in blood DNA and were therefore true sporadic cases. In 175 cases an RB1 mutation was detected, identifying these patients as heritable cases. When screening parents and, if available, relatives of these 175 patients, 12 cases were detected in which one parent also carried the mutation but did not develop retinoblastoma. Based on this data, the authors strongly recommend screening of retinoblastoma patient’s parents and relatives irrespective of clinical status or family history.

In total, the authors identified 14 patients with one asymptomatic parent and considered this to be cases caused by low penetrance variants of RB1. When analyzing the parental origin of these variants, 9 were transmitted from fathers, 5 from mothers. However, upon closer examination, three of the mothers showed retinomas, considered as benign form of retinoblastoma. These three mothers were therefore not truly asymptotmatic. One other mother was excluded because of mosaic presentation of the RB1 variant. The parental cause of low penetrance RB1 mutations was therefore corrected to 9 paternal to 1 maternal. The authors concluded “that parental origin of an RB1 mutation defines penetrance of hereditary retinoblastoma”.

The findings presented agree with other studies conducted by other centers in showing a relatively high rate of constitutional RB1 mutational variants in patients without family history. Also, the analysis of parental transmission of low penetrance variants is of value and is in line with previous studies showing a bias towards paternal transmission of such variants.

Points of criticism:

- Materials and Methods: information about patient consent and ethical review are missing and have to be provided.

- Materials and Methods: information about statstical methods are missing and have to be provided.

- Use of numbers of families and numbers of patients appear to be used equivalent - but it appears that some families have more than one patient. This should be considered and wording should be reviewed and maybe adjusted to avoid misunderstandings.

- Simple summary/abstract: The statement “... that parental origin of an RB1 mutation defines penetrance of hereditary retinoblastoma.” is not correct. Most variants of RB1 show full penetrance and can be transmitted by either parent. Although even for full penetrance variants a bias towards paternal inheritance has been described. It is rather that parental origin of a low penetrance RB1 mutation influences the likelihood of developing retinoblastoma. This statement should be changed.

- The pairs of terms “sporadic - familiar” (based on family history no/yes) and “non-hereditary - hereditary” (based on absence or presence of constitutional RB1 mutation) should be taken into account and should be used consistently throughout the text.

- The authors do not comment on mutation data of tumor material and if these would correlate with the mutations found in blood. If tumor data are available, this information should be included.

- Line 217: “with or without a family history” should be changed to “with a family history”.

- Line 218: incomplete penetrance is mentioned in 16 families (11 paternal/5 maternal) - Fig. 2 relates to 14 cases/patients in total (9 paternal/5 maternal) - this is somewhat inconsistent and should be clarified.

- Discussion line 277ff: downstream initiation at an alternative ATG for RB1 protein translation should also be considered as possible mechanism of low penetrance effect of exon1 mutations. For reference see publications of the Mittnacht group, for example PMID: 16988938.

- Discussion line 293: HUGO nomenclature of KIAA0649 should be used, change to PPP1R26P1

- Reference should be made, which of the identified low penetrance mutations were already described as such (for example c.607+1G>T and c.1981C>T).

Author Response

Dear reviewer,

Thank you for your general positive perception of our manuscript, and for your criticism.

We have addressed your points of criticism as follows:

- Materials and Methods: information about patient consent and ethical review are missing and have to be provided.

We have added this information to Materials and Methods. Informed consent form and a copy of the Institutional Ethics Committee protocol were sent to the Editor upon manuscript submission.

- Materials and Methods: information about statstical methods are missing and have to be provided.

We have added this information to Materials and Methods.

- Use of numbers of families and numbers of patients appear to be used equivalent - but it appears that some families have more than one patient. This should be considered and wording should be reviewed and maybe adjusted to avoid misunderstandings.

In this very study, it is a specific low penetrance mutation in the family we are interested in, whatever the family size. Therefore, it seems correct to equate the number of patients with the number of families.

- Simple summary/abstract: The statement “... that parental origin of an RB1 mutation defines penetrance of hereditary retinoblastoma.” is not correct. Most variants of RB1 show full penetrance and can be transmitted by either parent. Although even for full penetrance variants a bias towards paternal inheritance has been described. It is rather that parental origin of a low penetrance RB1 mutation influences the likelihood of developing retinoblastoma. This statement should be changed.

We have changed this statement in simple summary/abstract accordingly.

- The pairs of terms “sporadic - familiar” (based on family history no/yes) and “non-hereditary - hereditary” (based on absence or presence of constitutional RB1 mutation) should be taken into account and should be used consistently throughout the text.

We have addressed this issue. In some cases, we chose to use “noninherited” instead of “non-hereditary”.

- The authors do not comment on mutation data of tumor material and if these would correlate with the mutations found in blood. If tumor data are available, this information should be included.

In Russia, conservative treatment of retinoblastoma is widely available, thus design of a present study was based on molecular testing of blood samples only. We have discussed appropriateness of such design for this study in the Discussion. It is also indicated on the flowchart of the study. In order to make it clearer from the beginning, we have now added this to the “Clinical samples” subsection of Materials and Methods.

- Line 217: “with or without a family history” should be changed to “with a family history”.

This line refers to low penetrance mutations only, and they were actually found in families both with and without a family history. We have changed the sentence as follows: “A total of 15 inherited low penetrance mutations were observed in our cohort of retinoblastoma patients”.

- Line 218: incomplete penetrance is mentioned in 16 families (11 paternal/5 maternal) - Fig. 2 relates to 14 cases/patients in total (9 paternal/5 maternal) - this is somewhat inconsistent and should be clarified.

There is no inconsistency between the text and the figure. At the bottom of figure 2 we demonstrate 9 paternal/5 maternal cases of mutation inheritance from unaffected parents, and there is a box to the right under “Penetrance and expressivity” showing two cases of bilateral retinoblastoma inherited from fathers with unilateral form. This makes 11 fathers who were either asymptomatic carriers or had a milder disease form then in children. To make it more visible, we have changed the text in this box from “Variable expressivity (unilateral RB in parents, bilateral in children)” to “Variable expressivity (unilateral RB in fathers, bilateral in children)”, and colored the box blue.

- Discussion line 277ff: downstream initiation at an alternative ATG for RB1 protein translation should also be considered as possible mechanism of low penetrance effect of exon1 mutations. For reference see publications of the Mittnacht group, for example PMID: 16988938.

Thank you for this, we have added the reference and the description of this mechanism to the Discussion.

- Discussion line 293: HUGO nomenclature of KIAA0649 should be used, change to PPP1R26P1

Fixed.

- Reference should be made, which of the identified low penetrance mutations were already described as such (for example c.607+1G>T and c.1981C>T).

We have added this to Conclusions: “…three previously reported ones, с.607+1G˃T [14, 22, 27], с.1981C˃T [12], and c.939G˃A [30]”.